# Nasal Floor Elevation—An Option of Premaxilla Augmentation: A Case Report

Ante Jordan [1], Marko Vuletić [2,3,*] , Mato Sušić [2,3], Luka Stojić [4] and Dragana Gabrić [2,3]

1   Department of Oral Surgery, Dental Polyclinic Zagreb, 10000 Zagreb, Croatia
2   Department of Oral Surgery, School of Dental Medicine, University of Zagreb, 10000 Zagreb, Croatia
3   Department of Dental Medicine, University Hospital Centre Zagreb, 10000 Zagreb, Croatia
4   Private Dental Clinic, 10000 Zagreb, Croatia
*   Correspondence: mvuletic@sfzg.hr

**Abstract:** The atrophic edentulous maxilla is demanding for dental implant placement because of extensive resorption of the alveolar ridge after teeth loss and, consequently, the proximity of the anatomical structures, nasal cavity, and maxillary sinus. Treatment options are short implants, guided bone regeneration, onlay grafts, Le Fort I osteotomy with interpositional bone grafting, distraction osteogenesis, or nasal floor elevation. Nasal floor elevation is a method of augmentation of premaxilla by raising the base of the nose. The aim of this case report is to evaluate the success of implants placed after nasal floor elevation. A 75-year-old female patient came to the Clinical Department of Oral Surgery, University Hospital Centre Zagreb, unsatisfied with her complete removable denture. Clinical and radiological examination revealed severe maxillary alveolar ridge atrophy. Nasal floor elevation was made under local anesthesia through aperture piriformis and lateral window in the distal part. After eight months, four implants were placed and, after period of osseointegration, a bar-retained implant overdenture was made. This case report shows that nasal floor augmentation can be considered among the surgical techniques to allow implant-supported rehabilitation of the atrophic anterior maxilla.

**Keywords:** nasal cavity; maxilla; bone grafting; dental implants; tantalum; osseointegration; implant-supported denture





## 1. Introduction

The loss of teeth affects the aesthetics and function of the orofacial region and consequently compromises the patient's quality of life [1–3]. The goal of modern dentistry is to restore oral function, appearance, and aesthetics and to improve patients' health.

Patients with an edentulous maxilla can be treated with conventional removable dentures or implant therapy. Implant prosthetic options can be fixed or removable implant-supported prostheses [3].

Implant placement in the maxilla is often limited by insufficient bone width and height after teeth loss and by the proximity of the anatomical structures, nasal cavity, and maxillary sinus. In such cases, clinicians may consider placing short implants or increasing the bone volume using one of the techniques such as guided bone regeneration (GBR), autogenous bone block grafting, Le Fort I osteotomy with interpositional bone grafting, distraction osteogenesis, or nasal floor elevation (NFE) [4]. Aside from the patient's general health and medical history, the choice of the technique mainly depends on bone quality, the extent of bone defect, and jaws' relationship [5].

NFE is a surgical procedure based on lifting the nasal mucosa and the augmentation of different types of grafting materials, allowing the placement of dental implants. NFE technique and its modifications offer excellent clinical results and avoid more complex regenerative procedures in cases of severe atrophy in the anterior maxilla when placing

dental implants [6–8]. The technique is not recommended in cases of recurrent epistaxis, previous septum repair, chronic recurrent rhinitis, or chronic known allergy [9]. The graft should not exceed 6 or 7 mm in height to avoid interference with the inferior concha [10].

Despite the fact that NFE was first described more than three decades ago, the literature describing the technique and the results obtained from its use is scarce [8–10].

After nasal floor augmentation, standard-length or short implants can be placed simultaneously or after graft healing as a two-stage approach [6–11].

Although implant dentistry is constantly evolving and efforts are being made to improve existing and to find new materials, titanium and its alloys are still the most commonly used material for the fabrication of dental implants [12].

Tantalum and its derivatives have been demonstrated to be promising biomaterials owing to their characteristics such as a high degree of osseointegration, biocompatibility, lower cytotoxicity, and higher corrosion resistance than titanium [13].

Tantalum, a highly biocompatible and corrosion-resistant metal, has been successfully used in orthopedics and dentistry, but the high elastic modulus, low porosity, and high cost of tantalum metal limit its clinical application [12].

In recent years, the emergence of porous tantalum, also known as tantalum trabecular metal, has solved these problems because of its elastic modulus and porous structure, which is similar to that of cancellous bone [12].

To our knowledge, no case report in the literature described nasal floor elevation and placement of dental implants containing tantalum.

The aim of this case report is to evaluate clinical outcomes of nasal floor augmentation using bovine bone grafting material with platelets rich in growth factors (PRGF) as well as the success of the tantalum dental implants placed in the grafted anterior maxilla.

## 2. Case Report

A 75-year-old female patient came to the Clinical Department of Oral Surgery, University Hospital Centre Zagreb, having been referred by a prosthodontist due to an inability to function with an upper complete removable denture. Her medical history was unremarkable, with no history of tobacco use. Clinical examination revealed edentulous maxilla, severe residual ridge resorption, and a shallow vestibule. Four mini-implants with ball attachment supported an overdenture in the mandible. Radiologic analysis using a panoramic radiograph and cone beam computed tomography (CBCT) scan revealed severe atrophy of the maxillary alveolar ridge and extensive maxillary sinus pneumatization (Figure 1).

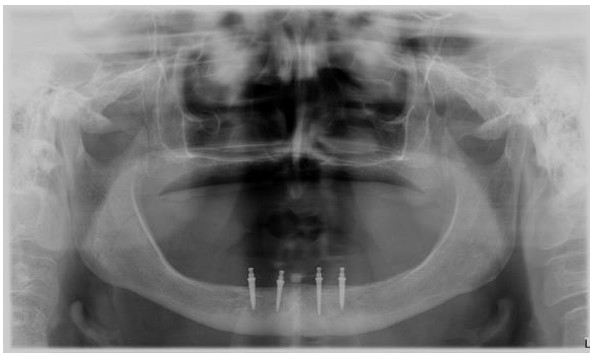

**Figure 1.** Preoperative panoramic radiograph.

After consultation with a prosthodontist, the treatment plan was nasal floor augmentation and placement of four dental implants for a bar-retained overdenture. The patient didn't have any medical or surgical contraindication to NFE and dental-implant placement. The patient received information concerning the present state and treatment plan and, in accordance with the ethical protocol of the Clinical Department of Oral Surgery, University Hospital Centre Zagreb, written consent was obtained.

Antibiotic therapy (Klavocin bid 875 mg amoxicillin + 125 mg clavulanic acid, Pliva, Zagreb, Croatia), was initiated twice daily for one day before surgery and then continued postoperatively for six days.

To obtain anesthesia of the surgical field, the bilateral infraorbital nerve block and nasopalatine nerve block by injection of a total of 6.8 mL of articaine hydrochloride 4% with adrenaline 1:200,000 (Ubistesin, 3M Deutschland GmbH, Neuss, Germany) were used. A crestal incision and two vertical releasing incisions were made in the anterior maxilla and the full mucoperiosteal flap was raised to expose the residual alveolar ridge, the anterior nasal spine, and the inferior and lateral piriform rim.

The access windows were created on the vestibular bone, one on each side of the inferior piriform rim, using the contra-angle handpiece (W&H, Bürmoos, Austria) and round diamond bur (Komet, Lemgo, Germany) under sterile saline irrigation (Figure 2a). After the removal of the bone window, the nasal mucosa was carefully lifted with manual elevators via lateral windows and piriform aperture (Figure 2b).

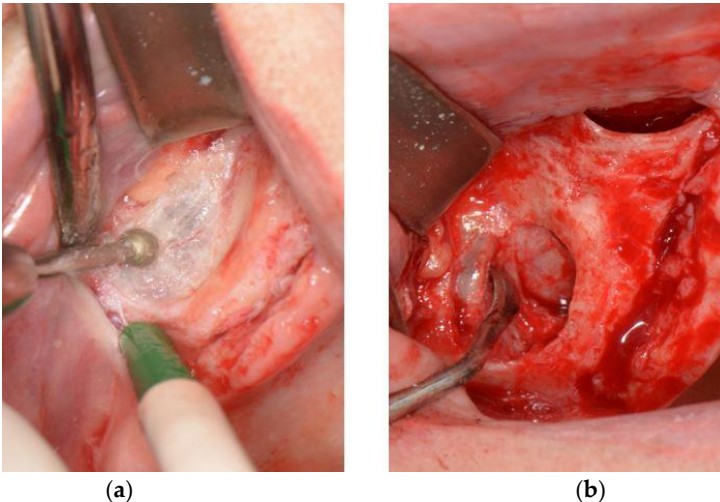

(**a**)          (**b**)

**Figure 2.** (**a**) Preparation of access window on the vestibular bone; (**b**) Detachment and elevation of the nasal mucosa.

PRGF (Endoret, BTI, Vitoria-Gasteiz, Spain) fraction F2 and PRGF membrane fraction F1 were prepared according to the manufacturer's procedure. The space under the elevated mucosa was filled with bovine bone grafting material (Cerabone, Botiss Dental, Berlin, Germany) mixed with PRGF fraction F2, and both lateral windows were covered with PRGF membrane fraction F1 (Figure 3).

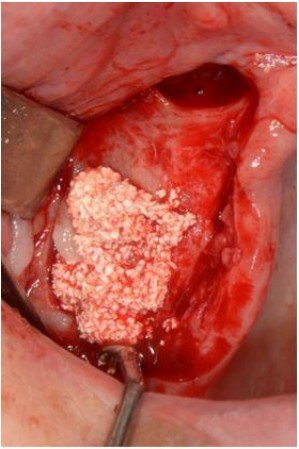

**Figure 3.** Bovine bone graft mixed with PRGF packed under the elevated nasal mucosa.

The mucoperiosteal flap was repositioned and simple interrupted sutures were placed for primary intention healing. To prevent early postoperative complications, analgesics and 0.12% chlorhexidine oral rinse were prescribed and the patient was given information on postoperative care. The sutures were removed 10 days after surgery. No postoperative complications occurred and the healing process was uneventful (Figure 4).

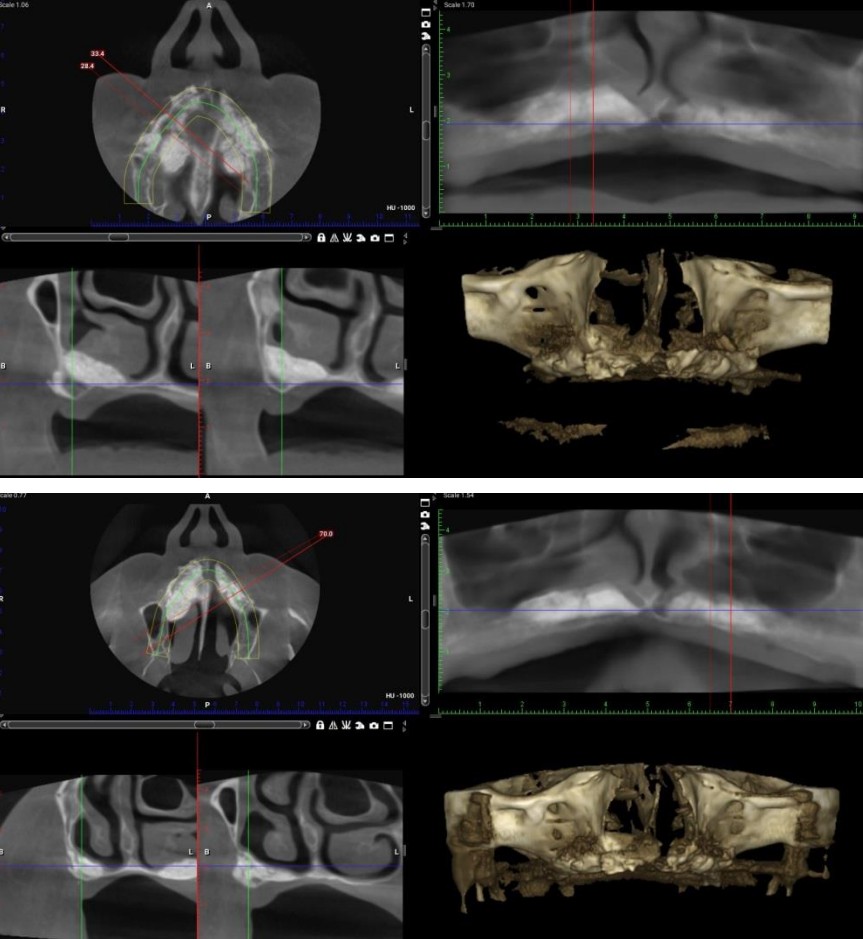

**Figure 4.** Postoperative control CBCT scan. A—anterior, P—posterior, R—right, L—left or lingual, B—buccal.

Eight months after nasal floor augmentation, the second surgical procedure—dental implant placement—was planned. After local infiltration of articaine hydrochloride 4% with adrenaline 1:200,000 into the anterior maxilla, a crestal incision was made and full-thickness envelope flap was elevated. The implant sites were prepared according to the procedure recommended by the implant manufacturer. Four implants (Trabecular Metal™ Dental Implants; Zimmer Biomet, Palm Beach Gardens, FL, USA) (3.7 × 8 mm) were placed in the anterior maxilla in the positions of canines and first incisors (Figure 5).

The flap was repositioned and sutured with simple interrupted sutures. Wound healing was uneventful, and the sutures were removed 10 days postoperatively.

The patient underwent postoperative radiological analysis two months after implant placement, revealing sufficient bone around the implants. The implants were uncovered and healing abutments were attached. Two weeks later, dental impressions were made to fabricate upper bar-retained overdenture (Figure 6).

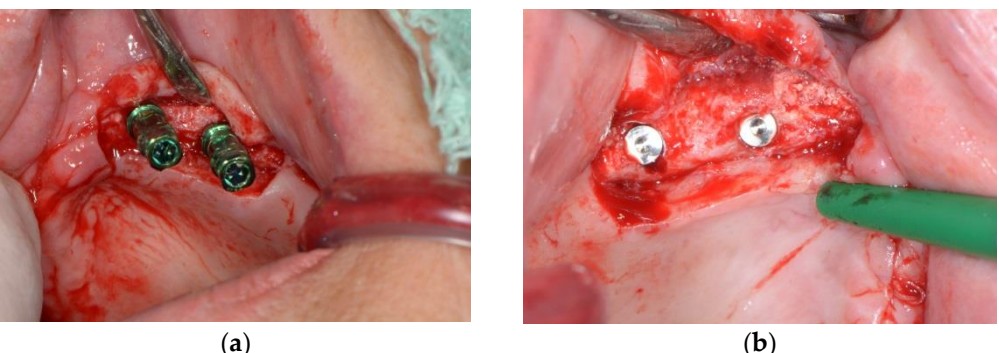

**Figure 5.** (**a**) Implant placement with transfer on left side of maxilla; (**b**) Implant placement with cover screw on right side of maxilla.

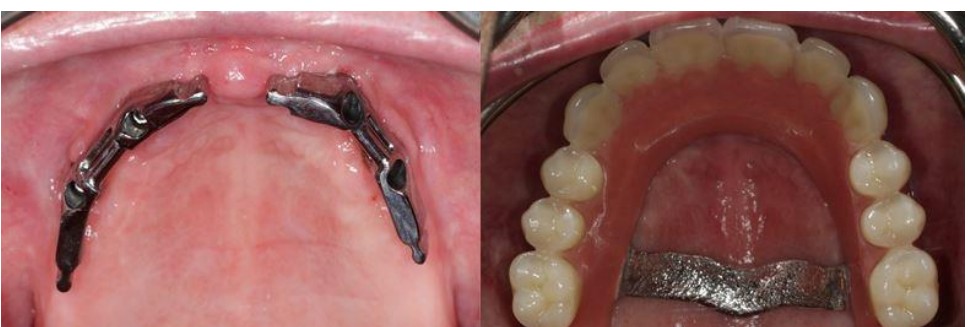

**Figure 6.** Bar-retained overdenture.

During the two-year follow-up period, neither mechanical nor biological complications were observed (Figure 7).

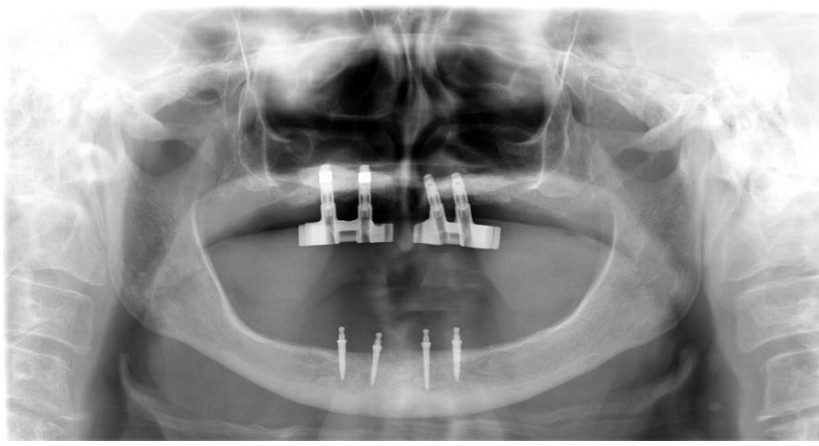

**Figure 7.** Panoramic image after 2 years of follow up.

## 3. Discussion

Despite the development of curative and preventive dental care in the last decades, edentulism continues to be a challenging problem for healthcare providers [14]. Prosthetic rehabilitation of the edentulous maxilla may be highly challenging because of the high aesthetic, functional, and biological demands [3,4].

Our patient was unsatisfied with her conventional complete denture, even though the treatment was performed by experienced prosthodontists. Her main complaint was difficulty in food-chewing and speaking, which may be due to severe maxillary residual ridge resorption and a shallow vestibule which consequently reduced the denture supporting area.

We proposed nasal floor elevation, implant placement, and an implant-supported overdenture as an alternative to a conventional complete denture, since studies showed that implant-supported overdentures improve retention and stability compared with conventional dentures [15,16], hence would improve oral health-related quality of life and patient satisfaction [15,16].

Several bone augmentation techniques in the atrophic anterior maxilla have been proposed in the literature as predictable and with high implant survival rates [6–8,17,18].

In our case, we recommended nasal floor augmentation as the optimal treatment due to the patient's age and because we wanted to avoid more complex procedures such as onlay bone grafting or interpositional bone grafting through a Le Fort I osteotomy. Reconstruction of the anterior maxillary atrophy with nasal floor elevation and dental implants is a minimally invasive treatment option in comparison with onlay grafting operations or Le Fort I surgery. This technique can be performed as office-based oral surgery under local anesthesia, with reduced postoperative consequences, and is cost-effective.

A variety of grafting materials have been used for nasal floor augmentation including autogenous bone, allografts, and xenografts alone or in combination, with high success rates [6,8,11]. In the present case, we used bovine bone grafting material because we wanted to avoid the risk of donor-site morbidity due to the patient's age. Considering severe resorption, we estimated that a sufficient amount of autogenous bone could not be harvested intraorally.

We mixed bovine bone grafting material with PRGF, since it's been described as improving handling and placement of xenograft when combined with PRGF and as decreasing pain and swelling for the patient [19].

It can be assumed that the fact that the patient is a non-smoker contributed to the success of the healing and implant osseointegration, since Garcia-Denche et al. reported in their study that smoking was the main reason for implant failure after nasal floor elevation [7].

In the available literature, it can be found that the procedure is not recommended in cases of recurrent epistaxis, previous septum repair, chronic recurrent rhinitis, or chronic known allergy [9]. In addition to this, our opinion is that the NFE procedure should be avoided in patients with compromised general health, uncontrolled systemic diseases, radiation therapy including the maxilla, ongoing chemotherapy, and mental disorders. The graft should not exceed 6 or 7 mm in height to avoid interference with the inferior concha [10] and normal function of the nasal cavity.

Nasal floor elevation can be done either in a single stage with simultaneous implant placement or in two stages with delayed implant placement. The long-term success of this technique has been reported both with the single-stage and two-stage approaches [6,11]. Although the two-stage approach has some disadvantages, such as a prolonged treatment period, increased risk of infection, and higher cost [20], since in our case we didn't have an adequate amount of crestal bone for primary implant stabilization, implants were placed eight months after graft healing.

In our case, the implants that we placed have a threaded titanium surface in the cervical and apical sections and in the middle section, and the internal titanium core is covered by a porous trabecular metal sleeve. Pores within the tantalum material provide significantly more surface area than conventional textured titanium implants and allow bone formation (ingrowth) within the tantalum microstructure—a phenomenon termed osseoincorporation [21]. Due to combining bone ongrowth with bone ingrowth, this type of implant may be indicated in poor healing situations, immediate or early loading of implants, and missing osseous structures requiring simultaneous implant placement and bone grafting [22].

Due to good implant stability 2 months after implantation and the lack of signs of bone-loss after functional loading of implants on follow-up appointments, it could be concluded that the implants had been successfully osseointegrated.

The limitation of this case report is the relatively short follow-up period compared with previously published studies [6,8].

## 4. Conclusions

This case report confirms that nasal floor augmentation using bovine bone grafting material with PRGF and placement of implants containing tantalum can be considered as a predictable technique for rehabilitation in the atrophic anterior maxilla. However, further studies and longer follow-up periods are required to confirm the findings of this case report.

**Author Contributions:** Conceptualization, A.J. and M.V.; methodology, A.J. and M.S.; software, L.S.; validation, A.J., M.V. and L.S.; formal analysis, M.S.; investigation, M.V.; resources, D.G.; data curation, L.S.; writing—original draft preparation, A.J.; writing—review and editing, A.J. and M.V.; visualization, M.S. and L.S.; supervision, D.G.; project administration, D.G. and M.S.; funding acquisition, D.G. All authors have read and agreed to the published version of the manuscript.

**Funding:** This research received no external funding.

**Institutional Review Board Statement:** Ethical review and approval were waived for this study due to retrospective reporting of a conventionally diagnosed and treated case.

**Informed Consent Statement:** Informed consent was obtained from all subjects involved in the study.

**Data Availability Statement:** Not applicable.

**Conflicts of Interest:** The authors declare no conflict of interest.

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
