# Peer review of "Nasal Floor Elevation—An Option of Premaxilla Augmentation: A Case Report"

_2673-4095, doi:10.3390/surgeries3040033_

Round 1
Reviewer 1 Report
I think this is a very interesting study. However, with a few modifications, it seems to be a better study.
1. It would be better to replace the CBCT picture in Figure 4 with a clearer picture.
2. Fig (a ) in 139 --> (a) needs to be modified
Author Response
Dear Reviewer,
After receiving the revision of our manuscript, we thank You for constructive comments and suggestions. We agree with your suggestions so we have revised the manuscript accordingly. All changes in the revised manuscript have been made and highlighted in red. Reponses to Your comments are given below.
Reviewer 1: I think this is a very interesting study. However, with a few modifications, it seems to be a better study.
Comment 1: It would be better to replace the CBCT picture in Figure 4 with a clearer picture.
Response: CBCT picture was replaced with two clearer picture of left and right side of maxilla.
Comment 2: Fig (a ) in 139 --> (a) needs to be modified.
Response: Figures 5a and 5b were replaced with better pictures.
Reviewer 2 Report
Dear Authors, globally the paper is a good case report but the English language is improvable, therefore I suggest a linguistic revision. Please correct the text following the comments in the modified PDF test file. A final X-ray or CBCT showing the inserted implants in the pristine bone and in the augmented area is highly recommended.
Author Response
Dear Reviewer,
After receiving the revision of our manuscript, we thank You for constructive comments and suggestions. We agree with your suggestions so we have revised the manuscript accordingly. All changes in the revised manuscript have been made and highlighted in red. Reponses to Your comments are given below.
Reviewer 2: Dear Authors, globally the paper is a good case report but the English language is improvable, therefore I suggest a linguistic revision. Please correct the text following the comments in the modified PDF test file.
Response: Unfortunately, we did not find yours comments in PDF file, but whole manuscript was revised by native English speaker.
Comment 1: A final X-ray or CBCT showing the inserted implants in the pristine bone and in the augmented area is highly recommended.
Response: Final panoramic image after 2 years of follow up was added as figure 7 in the manuscript.
Reviewer 3 Report
Abstract
“The aim of this case report was to evaluate the success of implants containing tantalum placed after nasal floor elevation”. A case report (or a case series) can’t evaluate the influence of tantalum implants on the final outcomes, as there is no control group to compare with. You can simply evaluate the success of implants inserted after nasal floor elevation, without postulating that tantalum may play a significant role.
Please change the sentence “The aim of this case report was to evaluate the success of implants containing tantalum placed after nasal floor elevation” into “The aim of this case report was to evaluate the success of implants placed after nasal floor elevation” and the sentence “The presentation of this case report showed that nasal floor augmentation and placement of implants containing tantalum can be considered as predictable technique for rehabilitation in the atrophic anterior maxilla” into “This case report showed that nasal floor augmentation can be considered among the surgical techniques to allow implant-supported rehabilitation of the atrophic anterior maxilla”
Discussion
Underline also in the Discussion section the contraindications for the technique and the fact that the graft should not exceed 6 or 7 mm in height to avoid interference with the inferior concha
Author Response
Dear Reviewer,
After receiving the revision of our manuscript, we thank You for constructive comments and suggestions. We agree with your suggestions so we have revised the manuscript accordingly. All changes in the revised manuscript have been made and highlighted in red. Reponses to Your comments are given below.
Reviewer 3:
Comment 1: Abstract
“The aim of this case report was to evaluate the success of implants containing tantalum placed after nasal floor elevation”. A case report (or a case series) can’t evaluate the influence of tantalum implants on the final outcomes, as there is no control group to compare with. You can simply evaluate the success of implants inserted after nasal floor elevation, without postulating that tantalum may play a significant role.
Please change the sentence “The aim of this case report was to evaluate the success of implants containing tantalum placed after nasal floor elevation” into “The aim of this case report was to evaluate the success of implants placed after nasal floor elevation” and the sentence “The presentation of this case report showed that nasal floor augmentation and placement of implants containing tantalum can be considered as predictable technique for rehabilitation in the atrophic anterior maxilla” into “This case report showed that nasal floor augmentation can be considered among the surgical techniques to allow implant-supported rehabilitation of the atrophic anterior maxilla”
Response: Both suggested sentences were added in the abstract and it was modified according to the comments.
Comment 2: Discussion
Underline also in the Discussion section the contraindications for the technique and the fact that the graft should not exceed 6 or 7 mm in height to avoid interference with the inferior concha
Response: Section in the Discussion regarding contraindications was added and modified to the reviewer's comment and highlighted in red.